# Comparative Transcriptomic Analysis of Cerebellar Astrocytes across Developmental Stages and Brain Regions

**DOI:** 10.3390/ijms25021021

**Published:** 2024-01-13

**Authors:** Wookbong Kwon, Dong-Joo Choi, Kwanha Yu, Michael R. Williamson, Sanjana Murali, Yeunjung Ko, Junsung Woo, Benjamin Deneen

**Affiliations:** 1Center for Cell and Gene Therapy, Baylor College of Medicine, Houston, TX 77030, USA; 2Center for Cancer Neuroscience, Baylor College of Medicine, Houston, TX 77030, USA; 3Department of Neurosurgery, Baylor College of Medicine, Houston, TX 77030, USA; 4Program in Cancer Cell Biology, Baylor College of Medicine, Houston, TX 77030, USA; 5Program in Immunology and Microbiology, Baylor College of Medicine, Houston, TX 77030, USA

**Keywords:** astrocyte, transcriptome, cerebellum, postnatal development, pediatric brain tumors

## Abstract

Astrocytes are the most abundant glial cell type in the central nervous system, and they play a crucial role in normal brain function. While gliogenesis and glial differentiation occur during perinatal cerebellar development, the processes that occur during early postnatal development remain obscure. In this study, we conducted transcriptomic profiling of postnatal cerebellar astrocytes at postnatal days 1, 7, 14, and 28 (P1, P7, P14, and P28), identifying temporal-specific gene signatures at each specific time point. Comparing these profiles with region-specific astrocyte differentially expressed genes (DEGs) published for the cortex, hippocampus, and olfactory bulb revealed cerebellar-specific gene signature across these developmental timepoints. Moreover, we conducted a comparative analysis of cerebellar astrocyte gene signatures with gene lists from pediatric brain tumors of cerebellar origin, including ependymoma and medulloblastoma. Notably, genes downregulated at P14, such as Kif11 and HMGB2, exhibited significant enrichment across all pediatric brain tumor groups, suggesting the importance of astrocytic gene repression during cerebellar development to these tumor subtypes. Collectively, our studies describe gene expression patterns during cerebellar astrocyte development, with potential implications for pediatric tumors originating in the cerebellum.

## 1. Introduction

Astrocytes, with their intricate structural and physiological properties, play vital roles in brain function. They communicate with neurons, regulate synaptic transmission and synaptogenesis, buffer neurotransmitters, and support the blood–brain barrier [1,2,3]. Despite their involvement in nearly every facet of brain physiology, astrocytes have traditionally been viewed as a homogeneous population of supporting cells due to their inability to generate action potentials [4]. However, recent studies over the past decade have dispelled this assumption with significant strides in unraveling the molecular diversity of astrocytes [4,5]. Molecular probing of the astrocyte transcriptome, facilitated by the brain-wide astrocyte-specific marker Aldh1l1 [6], has revealed extensive molecular heterogeneity across brain regions [7,8,9,10]. Despite recent advances in our understanding of astrocyte diversity, the molecular heterogeneity of astrocytes during brain development remains incomplete. 

The cerebellum has long been recognized as the central area for motor coordination within the central nervous system (CNS). In addition, it is now known to be related to other various behavioral outputs, such as cognitive and emotional function [11,12,13]. During cerebellar development, essential biological processes, including proliferation, differentiation, migration, and synaptogenesis occur, establishing the cerebellar circuits for diverse behavioral outputs [11,14]. Dysregulation of these processes contributes to a range of neurological disorders, including autism, neurodevelopmental disorders, ataxia, and pediatric brain tumors originating in the cerebellum, such as ependymoma and medulloblastoma [11,15,16,17]. Roles for diverse astrocyte populations in these various cerebellar functions remain obscure.

Glial cells in the cerebellum control various aspects of neuronal circuits, such as regulating the excitatory–inhibitory balance, responding to nearby neurotransmitters, or maintaining connections among cerebellar networks [18,19,20]. First described and classified by Ramón y Cajal based on morphology and position, glial cells in the cerebellum are now categorized into three major types: (1) Bergmann glia (BG), which align with Purkinje cell (PC) somata to organize the PC layer and surround granule cell (GC) axons, PC dendrites, and synapses in the molecular layer, (2) Velate astrocytes (VA), with bushy, star-shaped processes located in the granular layer, and (3) fibrous astrocytes (FA), characteristic of the white matter [14,21]. Despite evidence that cerebellar astrocytes exhibit cellular heterogeneity, our understanding of their molecular heterogeneity remains poorly defined. To understand this in the context of cerebellar development, it is crucial to decipher the molecular signatures during the immediate postnatal period.

In this study, we performed transcriptomic profiling of postnatal cerebellar astrocytes using *Aldh1l1-GFP*-based RNA sequencing (RNA-Seq) and identified temporal-specific gene signatures during cerebellar development. We pinpointed cerebellar-specific, astrocyte-enriched genes at each time point by comparing them with region-specific astrocyte DEGs from the cortex, hippocampus, and olfactory bulb. In addition, we conducted a comparative analysis of cerebellar gene signatures with gene lists from pediatric brain tumors of cerebellar origin, which identified cerebellar-specific P14 downregulated genes that exhibited increased expression in pediatric tumors. Altogether, these findings reveal differential expression of transcriptomes in cerebellar-specific astrocytes during postnatal development compared with other brain regions, and transcriptomic relationships with cerebellar-specific pediatric cancers, medulloblastoma, and ependymoma.

## 2. Results

### 2.1. Dynamic Astrocyte Marker Expression Changes during Cerebellar Development

To understand how astrocyte expression patterns evolve during cerebellar postnatal development, we conducted co-immunostaining of green fluorescent protein (GFP) and Glial Fibrillary Acidic Protein (GFAP) in the cerebellum of *Aldh1l1-GFP* reporter mice at P7, P14, and P28 time points (Figure 1A). We observed robust changes in cerebellar layers during these initial stages of postnatal development, which were accompanied by alterations in Aldh1l1-GFP expression. Notably, the P7 cerebellar cortex exhibited a characteristic external granular layer (EGL) with a relatively unorganized Purkinje cell (PC) layer and granule cell (GC) layer (Figure 1Aa’). However, by 14, the molecular, PC, and GC layers were observed in the cerebellar cortex at P14 and P28 (Figure 1Ab’,c’). Additionally, we analyzed the area of Aldh1l1-GFP or GFAP from three major astrocyte layers: Bergmann glia (BG), Velate astrocytes (VA), and fibrous astrocyte (FA) (Figure 1B,C). During layer development of the cerebellum, three types of astrocytes exhibited distinct morphological changes along with variations in GFAP expression patterns. The initial GFP-positive region for all three astrocyte types was comparable at P7 but underwent rapid expansion in the BG layer, which exhibited a reduction in the VA layers and remained unchanged in the FA area. (Figure 1D–F). GFAP expression patterns, however, differed notably from GFP; FAs exhibited consistently high GFAP expression across all time points, whereas BGs and VAs demonstrated a gradual increase in GFAP expression throughout development (Figure 1G–I).

### 2.2. Stage-Specific Astrocytic Gene Expression Signatures during Postnatal Cerebellar Development

To evaluate how astrocytic gene expression changes during cerebellar development, we performed RNA sequencing on GFP-sorted cells from cerebellum of *Aldh1l1-GFP* mice at P1, P7, P14, and P28 time points (Figure 2A). As an unbiased approach for examining global gene expression patterns at each time point, we initially performed principal component analysis (PCA) using a rlog-transformed gene expression matrix of global gene expression > 1 for each time (PC1: 72% variance, PC2: 13% variance). The PCA revealed distinct gene expression patterns that were unique for each time point (Figure 2B). The time points P14 and P28 displayed the greatest variation in expression pattern when compared with P1 and P7, suggesting that the transcriptomes of astrocytes at P14 and P28 are markedly different from astrocytes at P1 and P7 (Figure 2B). From this dataset, we found that 1337 genes were differentially expressed from P1 to P7, while 2690 genes were differentially expressed from P7 to P14, and 2539 genes were differentially expressed from P14 to P28. To validate astrocyte-specific signatures in our molecular profile, we conducted a comparison between our dataset and existing gene signatures derived from adult cerebellar astrocytes [10]. This comparison confirmed progressive increases in the expression patterns of astrocytic signatures, which were aligned closely with those identified in adult cerebellar astrocytes (Figure 2C,D). 

Next, we conducted differential gene expression analysis by comparing the time point expression profiles to determine temporal-specific differentially expressed genes (DEGs). The identified DEGs (up: 186 genes, down: 280 genes at P1, up: 109 genes, down: 199 genes at P7, up: 136 genes, down: 58 genes at P14, and up: 345 genes, down: 1073 genes at P28, Appendix A) are unique to their respective time points, showing significant up- or downregulation and exhibiting a fold change >0.75 at *p* < 0.05. Figure 2E visualizes the expression of each set of DEGs across every time point, highlighting the enrichment of these genes within their temporal context. To decipher the cellular pathways regulated by these DEGs, we conducted biological Gene Ontology (GO) analysis on the DEGs from each time point (Figure 2F). Interestingly, we observed dynamic changes in gene expression patterns during cerebellar development, with DEGs related to axon guidance and projections enriched at P1 and downregulated at P28, while DEGs associated with myelination were downregulated at P1.

To gain additional insights into these astrocytic gene expression patterns, we sought to pinpoint the expression signatures that undergo up- or downregulation at specific time points during cerebellar development. We identified the expression of gene signatures that are specifically up- or downregulated from each time point compared with their preceding counterparts (Figure 2G, Appendix A). The identified DEGs exhibited significant up- or downregulation, with a fold change >0.75 at *p* < 0.05 compared with their respective prior time points (up from P7: 298 genes, up from P14: 840 genes, up from P28: 405 genes, down from P7: 324 genes, down from P14: 858 genes, and down from P28: 1278 genes). We then performed biological GO analysis with these DEGs, revealing widespread alterations in the GO biological processes at each time point (Figure 2H). For example, DEGs associated with synaptic and axon functions were broadly altered in both up- and downregulated genes at P7, while various lipid and peptide metabolic processes were altered at P14. We also found that genes related to vascular, gland, and immune functions were altered at P28.

### 2.3. Region-Specific Astrocyte-Enriched Gene Signatures during Postnatal Development

To investigate whether developing cerebellar astrocyte gene signatures are universal or region-specific, we conducted a comparative analysis by aligning cerebellar-enriched gene signatures with temporally corresponding DEGs from cortical, hippocampal, and olfactory astrocytes as previously reported datasets [22] (Figure 3A). Next, we performed PCA using a rlog-transformed gene expression matrix of global gene expression >1 for each region across four time points (PC1: 37% variance, PC2: 25% variance). The PCA applied to data encompassing all four regions across four time points revealed significant differences between cerebellar gene signatures and those from the other three brain regions (Figure 3B). In our analysis, we compared DEGs across all four regions at the same time points to identify universally conserved astrocyte genes. The results indicated minimal gene conservation during the P1 to P7 (12 shared genes, Figure 3C,D) and P14 to P28 periods (2 shared genes, Figure 3G,H), while a more substantial set of 214 shared DEGs emerged during the P7 to P14 period (Figure 3E,F). These results suggest that cerebellar astrocytes undergo distinct gene expression changes during the P1 to P7 and P14 to P28 periods, setting them apart from other brain regions. In addition, during the P7 to P14 period, these astrocytes express a subset of genes that are globally conserved (Figure 3I). To gain additional insight into these gene expression patterns, we conducted GO analysis using the enriched genes identified in the four brain regions at each time point (Figure 3J–L). We observed that DEGs in cerebellar astrocytes across three time points (P7, P14, and P28) are linked to nervous system development and the regulation of synaptic function. In contrast, DEGs in the other three regions (cortex, hippocampus, and olfactory) exhibited diverse functional GO terms associated with post-developmental processes. These results suggest that cerebellar astrocytes undergo distinct postnatal biological processes that set them apart from those observed in other brain regions.

### 2.4. Identification of Cerebellar-Specific Enriched Astrocyte Gene Signatures during Postnatal Development

We next sought to identify cerebellar-specific enriched genes by comparing the cerebellar-specific gene sets with those of the other three brain regions used for comparison (cortex, hippocampus, and olfactory) at the same time point. This analysis resulted in the identification of over 900 cerebellar-specific genes at each time point (Figure 4A). Through the GO analysis of these genes, we found distinct biological processes associated with the P1 and P7 gene signatures, while P14 and P28 shared several common biological GO terms, such as functions for synaptic transmission or calcium ion regulation. Specifically, P1-enriched astrocyte genes were associated with functions related to extracellular matrix organization, whereas P14-enriched genes were predominantly linked to cell cycle regulation (Figure 4B). To identify cerebellar-specific, astrocyte-enriched genes that are conserved, we compared cerebellar-specific enriched genes across the four time points (P1, P7, P14, and P28), resulting in the isolation of 73 genes (Figure 4C,D). We next compared these genes with the adult cerebellar-enriched astrocyte gene signature [10], leading to the identification of 41 developmental cerebellar-specific, astrocyte-enriched genes and 32 genes exhibiting expression into the adult stage (Figure 4E). GO analysis revealed that developmental cerebellar-specific, astrocyte-enriched genes are primarily associated with gene regulatory functions, including chromosome organization and telomere maintenance. In contrast, genes that maintain expression into the adult stage are linked to certain processes, such as neuronal differentiation, renal system development, and microtubule bundle formation (Figure 4F). Figure 4G displays a heatmap of developmental-specific astrocyte-enriched genes that consistently exhibit high expression in cerebellar astrocytes throughout postnatal development.

### 2.5. Identification of Cerebellar-Specific Gene Signatures Expressed in Pediatric Brain Tumors

The cerebellum is a frequent location for pediatric brain tumors, including ependymoma and medulloblastoma [23,24,25]; therefore, we next assessed the association between cerebellar-specific astrocyte gene signatures and genes highly enriched in pediatric brain tumors originating in the cerebellum. To achieve this, we employed gene lists associated with Posterior fossa subgroup A ependymoma (PFA-EPN), Supratentorial ependymoma (ST-EPN), Shh-Medulloblastoma (SHH-MB), and Group 3 and 4 Medulloblastomas (G3-MB and G4-MB) (Figure 5A) [23,26]. Next, we compared genes enriched in these five different tumor types with our temporal-specific gene signatures at P7, P14, and P28 (Figure 5B, Appendix A). Interestingly, all five types of tumors exhibited overlap with our cerebellum astrocyte signatures, highlighted by the downregulated DEGs at P14 with over 100 genes. Subsequently, we conducted GO analysis on genes from tumor groups that exhibited overlap with downregulated DEGs at P14, revealing an array of biological processes related to DNA regulatory functions, cell cycle processes, and chromosome regulation (Figure 5C,D). Next, we isolated 36 genes that are highly expressed in all five types of tumors (Figure 5E) and observed that the 36 cerebellar astrocyte genes downregulated at P14 exhibited their highest expression at P7 and underwent a significant decrease thereafter from P14 (Figure 5F). Comparative analysis of the four brain regions in our study at corresponding time points (P1, P7, P14, and P28) revealed that the expression of these genes is cerebellar-specific and temporally elevated at P7 (Figure 5G). Finally, we identified 16 cerebellar-specific enriched genes that exhibited a significant decrease from P14 (Figure 5H,I). To validate their expression patterns at each time point, we immunostained two candidate proteins, Kif11 and HMGB2, and observed that both proteins highly co-localize with *Aldh1l1-GFP* astrocytes. Moreover, their expression patterns were consistent with the temporal changes observed in our DEG analysis (Figure 5J and Appendix A). Taken together, these data support the notion that our bioinformatic analysis can identify stage-specific cerebellar astrocyte genes enriched in pediatric tumors of cerebellar origin. This discovery could serve as a starting point for understanding developmental astrocyte marker gene changes, potentially offering a prognostic feature for pediatric brain tumors of cerebellar origin.

## 3. Discussion

Although the molecular heterogeneity of astrocytes across various brain regions has been extensively profiled in recent years [7,8,9,10], a comprehensive examination of cerebellar astrocytes during postnatal development has not been conducted. In this study, we hypothesized that the differential expression of astrocyte gene signatures during postnatal development is a crucial factor in controlling cerebellar development and the acquisition of mature astrocytic properties. To investigate, we conducted *Aldh1l1-GFP*-based RNA sequencing across a series of postnatal time points, thus providing transcriptomic profiling of developing cerebellar astrocytes. A subsequent analysis of temporal-specific DEGs and region-specific DEGs revealed functionally redundant gene ontologies associated with the distinctive gene profiles from each time point or region. Finally, we identified temporally enriched cerebellar astrocyte genes in the early postnatal period that are highly enriched in pediatric brain tumors of cerebellar origin. These results reveal arrays of astrocyte gene signatures that are temporally unique during cerebellar development and highly enriched in pediatric brain tumors of cerebellar origin.

### 3.1. Distinct and Conserved Gene Signatures in the Transcriptome of Cerebellar Astrocytes during Development

Transcriptomic analyses across distinct brain regions have begun to dissect the relationship between region-specific astrocytic gene signatures and their contributions to the regulation of neuronal circuits and behaviors relevant to each specific brain region [27,28,29,30]. Nevertheless, questions persist regarding how regionally distinct astrocytes in the brain develop to exhibit these unique molecular and functional features. Several studies have demonstrated spatial patterning during spinal cord development, raising the question of how these mechanisms influence astrocyte heterogeneity in the adult brain [30,31]. These observations suggest that astrocytes may possess developmentally encoded molecular identities. Another possibility is that astrocyte heterogeneity is a product of interactions with neighboring neurons and occurs after specification and migration [28,32].

Building upon our observations from immunostaining developing cerebellar astrocytes (Figure 1) and the concurrent transcriptome analysis of differential gene expression (Figure 2), we identified significant structural and molecular changes during the transition phase from P7 to P14 postnatal time points. Structurally, the three major layers (molecular, PC, and GC layers) exhibited distinct and clear boundaries at P14 through P28. This was accompanied by significant changes in GFP and GFAP expression in Bergmann glia (BG) and Velate astrocyte (VA) layers. At a molecular level, cerebellar astrocytes underwent dramatic changes in both global and specific gene expression patterns during the P7 to P14 transition phase (Figure 2B and Figure 3F). GO analysis at P14 revealed upregulated genes associated with post-developmental GO terms, such as CNS myelination, axon ensheathment, or lipid metabolism, while downregulated genes at P14 were linked to processes like neuron migration, differentiation, or the Wnt signaling pathway, which are known to be crucial for cerebellar development (Figure 2F,H) [33]. Intriguingly, the P7 to P14 phase revealed a remarkable increase in the number of genes shared among astrocytes, which are globally expressed in other brain regions (Figure 3F). The conserved astrocytic genes identified at P14 are associated with various synaptic functions (Figure 3K), suggesting the initiation of the expression of functional genes associated with neuronal interactions at this time point. Considering the essential role of PCs in cerebellar synaptic circuitry, which involves integrating excitatory synaptic inputs from both climbing fibers and parallel fibers and sending solely inhibitory output [34,35], the data support the critical correlation between the key regulatory functions of astrocytes in early cerebellar circuitry formation. As an increasing number of studies highlight the connection between cerebellar glia and neurological dysfunction [15,18,20,34], and given that early symptoms of numerous cerebellar-related neurodevelopmental disorders emerge during the early postnatal stage [34,36,37], our data also support the notion that transcriptomic changes in cerebellar astrocytes play a significant role in the development of cerebellar circuits during the P7 to P14 period. To address the important question of how region-specific astrocytes exhibit differential transcription, additional studies are essential to identify the underlying mechanisms specific to these astrocytes. While we acknowledge the need for further investigation, we posit that this difference may be regulated by transcription factors within region-specific astrocytes or through interactions with neurons specific to those regions during development [38,39]. 

### 3.2. Temporal Cerebellar Astrocyte Gene Signature: Implications for Pediatric Brain Tumors of Cerebellar Origin

The cerebellum is a frequent location for pediatric brain tumors, including ependymoma and medulloblastoma [23,25,40]. Comparative analysis based on murine cerebellar atlases suggests that PFA-EPN may trace its roots to radial glial subtypes [41]. The origin of medulloblastomas, however, varies among subgroups. Studies indicate that GC precursors are the likely origin of SHH-MB [42,43], while G3-MB and G4-MB subtypes are believed to originate from the cerebellar rhombic lip subventricular zone, a progenitor area for glutamatergic unipolar brush cells [42,44]. The Wnt medulloblastoma subtype originates from a distinct brain region [45]. Despite these findings, malignant transformation of tumor cells alters their cellular properties and transcriptional profiles, thus complicating the identification of key factors for tumorigenesis specifically within the cerebellum. Our comparative analysis of temporal, cerebellar-specific astrocyte gene signatures and enriched gene lists from pediatric tumors of cerebellar origin suggests a potential link between astrocytic transcriptomic transitions and the gene signatures associated with pediatric brain tumors (Figure 5B). Astrocytic genes enriched in all five cerebellar-origin pediatric brain tumors were cerebellar-specific and exhibited elevated expression at P7 but experienced a significant decrease after P14. This observation suggests that the temporal transcriptomic dynamics of astrocytes during the transition from development to functionally active are crucial, irrespective of the cell of origin of cerebellar pediatric tumors.

## 4. Materials and Methods

### 4.1. Animals

All animals were treated in compliance with the US Department of Health and Human Services, NIH guidelines, and Baylor College of Medicine institutional animal care and use committee (IACUC) guidelines. Mice were housed in a 12 h light–dark cycle environment with food and water available. Both male and female mice were used for all experiments, and littermates were randomly allocated to experimental groups. All mice used in this study were maintained on the C57/BL6J background. For FACS purification and visualization of astrocytes, the *Aldh1l1-GFP* reporter mouse line was used [46].

### 4.2. Tissue Preparation

For immunostaining, mice were anesthetized and transcardially perfused, first with saline solution and then with 4% paraformaldehyde in 0.1 M phosphate buffer (PB; pH 7.2). Brains were stored at 4 °C in 4% paraformaldehyde for 2 d, and then in a 20% sucrose solution until they sank. Six separate series of 40 μm coronal brain sections were obtained using a Model CM3050S cryostat (Leica, Wetzlar, Germany) and stored in an anti-freeze stock solution (PB containing 30% glycerol and 30% ethylene glycol, pH 7.2) at 4 °C before use.

### 4.3. Immunofluorescent Staining

For immunofluorescence staining, every 6th serial section in each set (~8 sections) was collected and rinsed two times with PBS containing 0.2% Triton X-100 (PBST). After blocking non-specific binding by incubating with 10% goat serum in PBST, sections were incubated overnight at room temperature with primary antibodies (GFP (1:1000 dilution; abcam, ab13970), GFAP (1:1000 dilution; abcam, ab4674), HMGB2 (1:500 dilution; Proteintech, 14597-1-AP), and Kif11 (1:500 dilution; Proteintech, 23333-1-AP)). Immunoreactive proteins were visualized using Alexa-Fluor-488- or Alexa-Fluor-555-conjugated secondary antibodies (1:600 dilution; Invitrogen, Carlsbad, CA, USA). 

### 4.4. Confocal Imaging and Image Analysis

For whole cerebellar imaging, fluorescence images were acquired using the Zeiss LSM 880 laser-scanning confocal microscope with a ×20 objective tile imaging with a frame size of 1024 × 1024 and a bit depth of 12 (Zen3.1). For the layer-specific imaging, fluorescence images were taken with a ×40 oil immersion objective with frame size of 1024 × 1024. Images were imported to Imaris Bitplane software (ver. 9.2.1), and 3D surface renderings (Figure 1B,C) were performed with the Imaris Surface module by reconstructing color-coded surface images based on the surface area of each astrocyte type.

### 4.5. FACS Sorting

We collected cerebellum at precise time points (P1, P7, P14, and P28) from the mouse brain and dissociated them using a protocol described previously [7]. Dissociated astrocytes from each time point were gated using the BD FACSDiva software (ver. 9.0.1) and sorted using the BD FACSAira III system with a 100 µm nozzle. Around 95,000 GFP+ astrocytes were collected per 1.5 mL tube, which contained 650 µL of buffer RLT (Qiagen, 79216, Hilden, Germany) with 1% β-mercaptoethanol. Finally, each sample was vortexed and rapidly frozen on dry ice.

### 4.6. RNA Extraction and Library Preparation

For the whole transcriptomic RNA-Seq, RNA was extracted from pelleted cells using the RNeasy Micro Kit (74004, Qiagen). RNA integrity (RIN R 8.0) was confirmed using the High Sensitivity RNA Analysis Kit (DNF-472-0500, Agilent (formerly AATI)) on a 12-capillary fragment analyzer. cDNA synthesis and Illumina sequencing libraries with 8 bp single indices were constructed from 10 ng total RNA using the Trio RNASeq System (0507-96, NuGEN, Leeds, UK). The resulting libraries were validated using the Standard Sensitivity NGS Fragment Analysis Kit (DNF-473-0500, Agilent (formerly AATI)) on a 12-capillary fragment analyzer and quantified using the Quant-it dsDNA assay kit (Q33120). Equal concentrations (2 nM) of libraries were pooled and processed for paired-end (R1: 75, R2: 75) sequencing of approximately 40 million reads per sample using the High Output v2 kit (FC-404-2002, Illumina, San Diego, CA, USA) on the NextSeq550 system according to the manufacturer’s instructions. For scRNA-Seq, a single-cell gene expression library was prepared according to the Chromium Single Cell Gene Expression 3v3.1 kit (10× Genomics). In brief, single cells, reverse transcription reagents, gel beads containing barcoded oligonucleotides, and oil were loaded onto a Chromium controller (10× Genomics) to generate single-cell gel beads in emulsion, in which full-length cDNA was synthesized and barcoded for each single cell. Subsequently, the gel beads in emulsion were broken, and cDNA from each single cell was pooled. After clean-up using Dynabeads MyOne Silane Beads, the cDNA was amplified through PCR. The amplified product was fragmented to the optimal size before end-repair, A-tailing, and adaptor ligation.

### 4.7. RNA Sequencing

RNA sequencing was performed by Admera Health (South Plainfield, NJ, USA, https://www.admerahealth.com). The final libraries’ quantities were assessed using Qubit 2.0 (ThermoFisher, Waltham, MA, USA) and the QuantStudio ^®^ 5 System (Applied Biosystems, Foster City, CA, USA), and the quality was assessed using TapeStation D1000 ScreenTape (Agilent Technologies Inc., Santa Clara, CA, USA). Equimolar pooling of libraries was performed based on QC values and sequenced on an Illumina^®^ NovaSeq X Plus 10B (Illumina, San Diego, CA, USA) with a read length configuration of 150 PE for an estimated 2–2.5B PE reads per lane (1–1.25B reads each side). RNA-Seq data have been deposited at the NIH GEO database (GSE171590, access date: 30 June 2024).

### 4.8. Bioinformatics Analysis

Sequencing files from each flow cell lane were downloaded, and the resulting fastq files were merged. Quality control was performed using fastQC (version 0.10.1) and MultiQC (version 0.9) [47]. Reads were mapped to the mouse genome mm10 assembly using STAR (version 2.5.0a) [48]. In R (version 3.5.2), Bioconductor packages GenomicAlignments (version 1.16.0) and GenomicFeatures (version 1.32.2) were used to build count matrices [49]. UCSC transcripts were downloaded from Illumina iGenomes in GTF file format. Normalization and differential gene expression analysis were performed using DESeq2 (version 1.20.0) [50]. Cerebellar DEGs and values at each developmental time point and cerebellar-specific DEGs and values from regional comparison at each time point are listed in Appendix A. Gene expression heatmaps were generated using ComplexHeatmap (V2.0.0) [8]. Gene Ontologies were determined using Enrichr and visualized using GOplot (version 1.0.2). To analyze the data with other brain regions at indicated time points, we used our published dataset [22]. To analyze the data with adult cerebellum signature genes, we used a published dataset [10]. To analyze the data with the medulloblastoma and ependymoma gene set, we used a published dataset for medulloblastoma [23] and ependymoma [26].

### 4.9. Statistical Analysis

Sample sizes and statistical tests can be found in the accompanying figure legends. Offline analysis was performed using Image J and Prism software (version 9) and Microsoft Excel. The statistical significance of differences between two groups was determined using unpaired, two-tailed Student’s t-tests. For comparisons of multiple means, statistical significance was determined through one-way analysis of variance (ANOVA) followed by a Newman–Keuls post hoc test.

### 4.10. Data Availability

The datasets generated for this study can be found in the GEO repository, accession GSE252519. The source data are provided in this paper. All other data in this article are available from the corresponding author upon reasonable request.

## Figures and Tables

**Figure 1 ijms-25-01021-f001:**
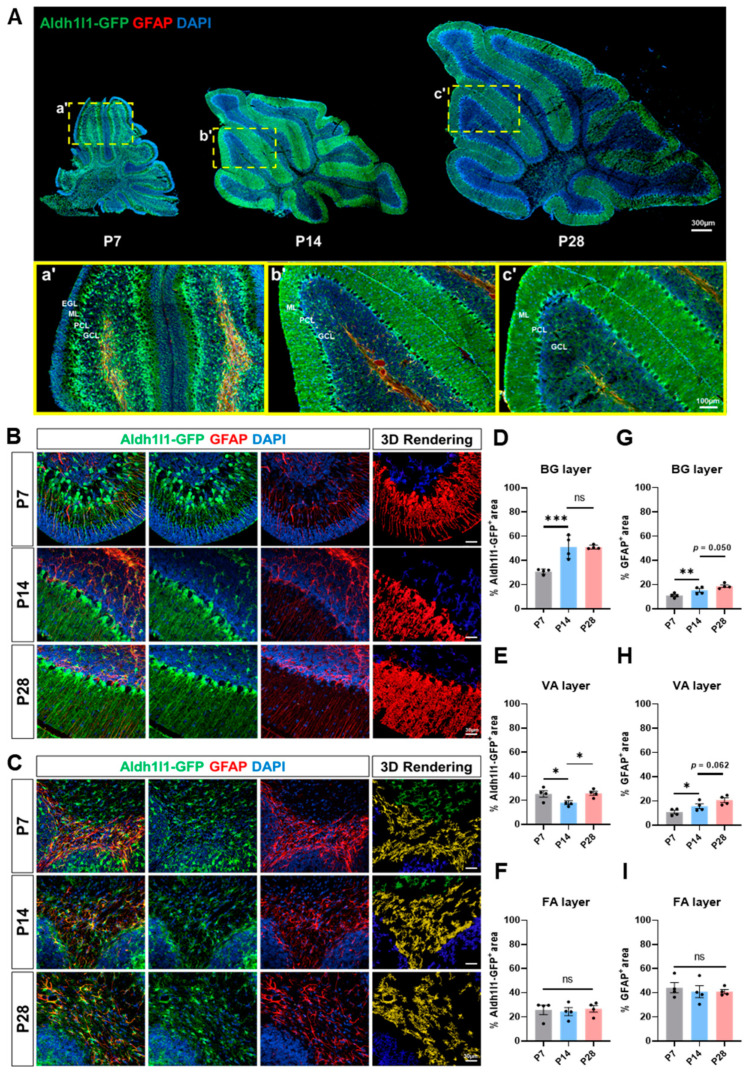
Temporal dynamics of astrocytic marker patterns coincide with cerebellar structural development. (**A**) Exemplar images of cerebellar sections from *Aldh1l1-GFP* mice immunostained for GFP (green) and Dapi (blue). Panels (**a’**–**c’**) (indicating yellow box) show higher magnitude images of each brain. (**B**) Representative images showing cerebellar cortex, including PC layer and molecular layer, where BGs and VAs are located. 3D rendering images illustrating the GFP-positive fractional area, with BGs highlighted in red and VAs in blue. (**C**) Representative images showing cerebellar white matter, where FAs are located. 3D rendering images illustrating the GFP-positive fractional area, with VAs highlighted in blue and FAs in yellow. (**D**–**F**) Graphs showing GFP-positive fractional area for BG, VA, and FA. (**G**–**I**) Graphs showing GFAP-positive fractional area for BG, VA, and FA (*n* = 4 for each group, ns (not significant), * *p* < 0.05, ** *p* < 0.001, *** *p* < 0.0001). Data are shown as the mean ± standard error of the mean (SEM).

**Figure 2 ijms-25-01021-f002:**
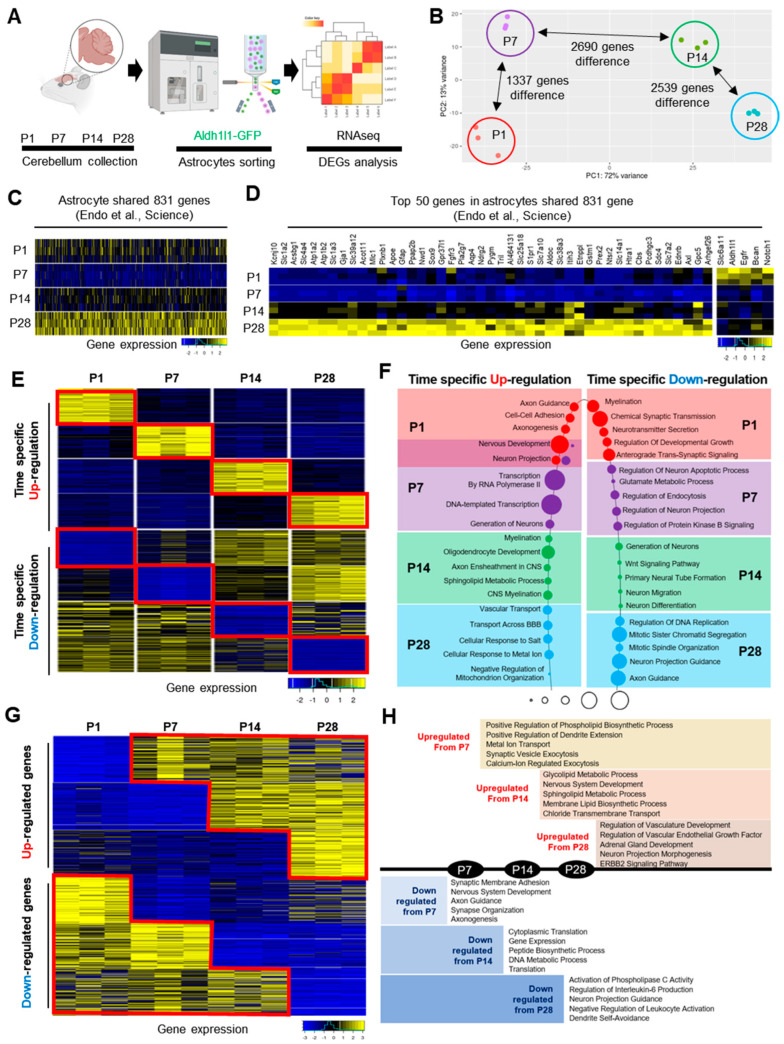
Identification of cerebellar astrocyte gene expression signatures during postnatal development. (**A**) Schematic of the approach used to investigate astrocyte transcriptome in mouse cerebellum. (**B**) Principal component analysis plot from RNA-Seq results of cerebellar astrocytes across time points. (**C**) Heat map showing gene expression changes of developing cerebellar astrocytes compared with adult astrocytes’ shared genes. (**D**) Heat map and gene lists of the top 50 genes from adult shared genes comparison [10]. (**E**) Heat map showing time-specific up- or downregulated genes across time points. (**F**) Top 5 biological GO terms of each time-specific up- or downregulated gene. (**G**) Heat map showing the genes up- or downregulated from each time point. (**H**) Top 5 biological GO terms of genes that were up- or downregulated from each time point.

**Figure 3 ijms-25-01021-f003:**
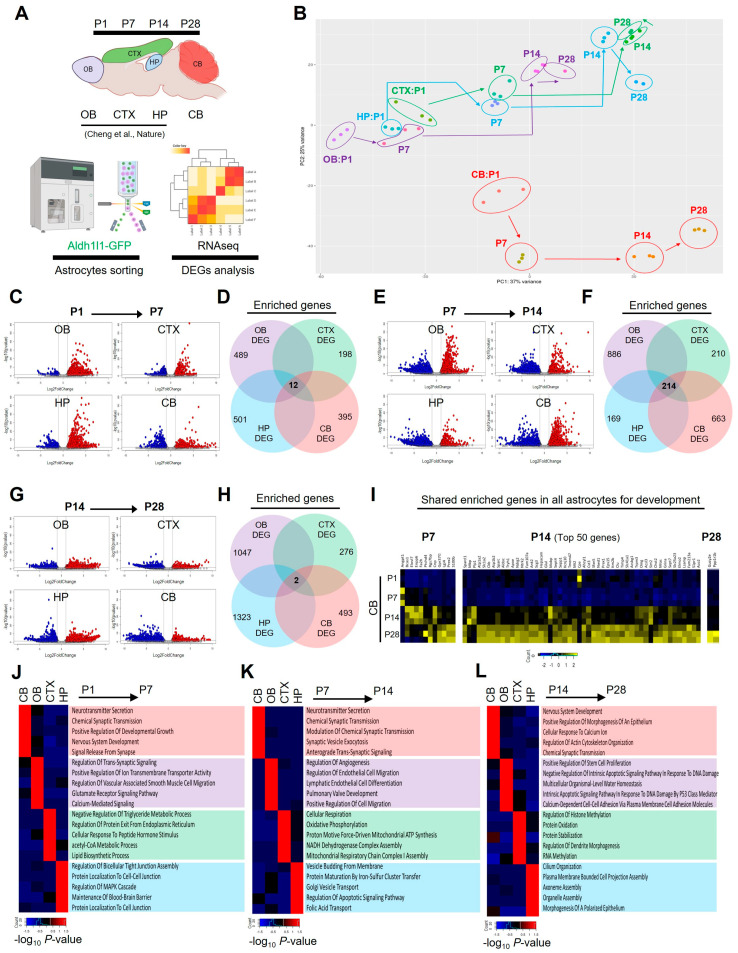
Enriched gene expression signature changes define region-specific astrocytes transcriptome during postnatal brain development. (**A**) Overview of the approach for investigating region-specific postnatal astrocyte transcriptome from different brain regions [22]. (**B**) Principal component analysis plot displaying RNA-Seq results of astrocytes from four brain regions across time points. (**C**,**E**,**G**), Heat maps showing DEGs of each time point. (**D**,**F**,**H**) Number of region-specific or globally conserved DEGs across the time points. DEGs of each time point are listed in Appendix A. (**I**) Heat map and gene lists showing globally conserved genes across the time points. (**J**–**L**) Top 5 biological GO terms of DEGs from four brain regions across the time points.

**Figure 4 ijms-25-01021-f004:**
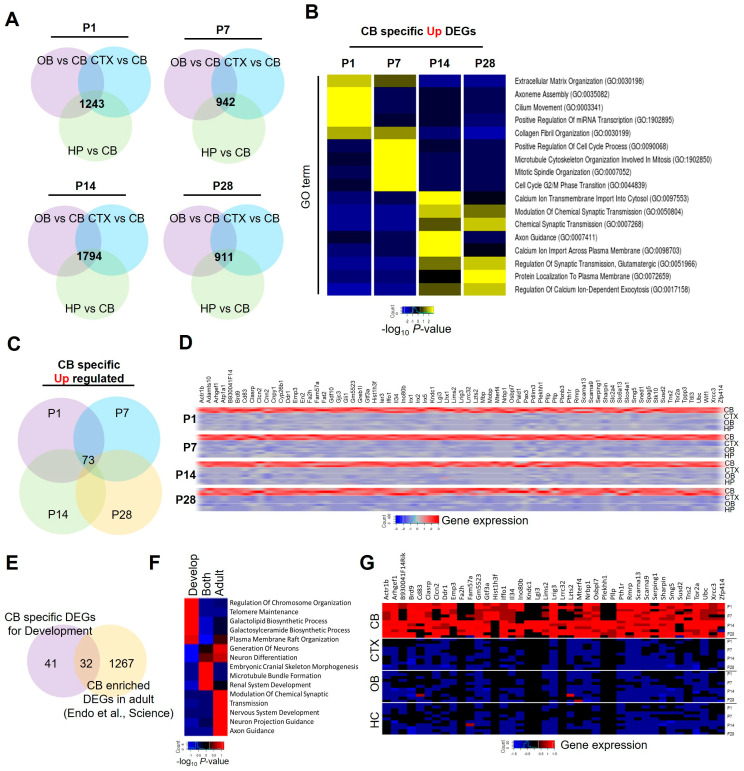
Isolation of CB-specific enriched astrocyte signature genes for postnatal development. (**A**) Venn diagrams showing cerebellar-specific genes at each time point. (**B**) Heat map and biological GO terms of cerebellar-specific enriched genes. (**C**) Venn diagram showing the number of genes conserved across the time points. (**D**) Heat map and gene list showing cerebellar-specific genes that are conserved across the time points. (**E**) Venn diagram displaying cerebellar-specific DEGs for development and cerebellar-enriched genes conserved in the adult stage [10]. (**F**) Biological GO terms and heat map of cerebellar-specific conserved genes. (**G**) Gene list and heat map of cerebellar-specific conserved genes.

**Figure 5 ijms-25-01021-f005:**
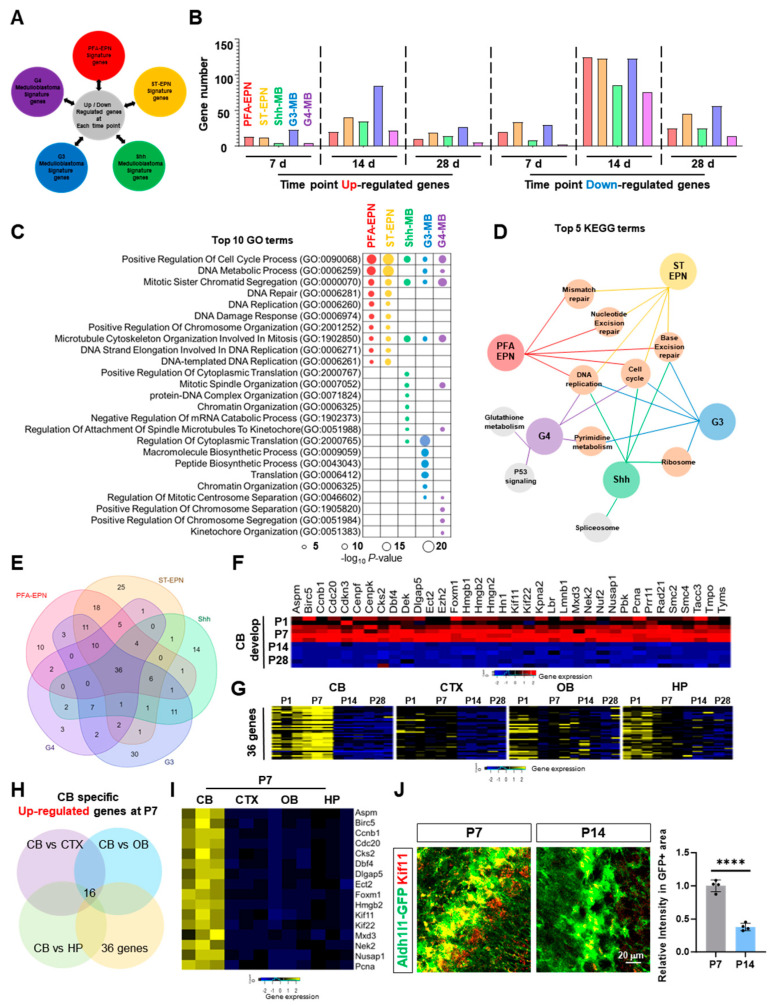
Identification of unique signature genes enriched in brain tumors of cerebellar origin (**A**) Schematics showing the types of brain tumors that were analyzed in this study. (**B**) Graph showing the number of genes from comparison analysis of up- or downregulated DEGs from each time point. (**C**) Top 10 biological GO terms from five types of brain tumors of cerebellar origin. (**D**) Top 5 KEGG terms of five brain tumor types. (**E**) Venn diagram displaying the shared or unique gene number from each tumor type. (**F**) Gene list of cerebellar brain tumor shared genes and heat map showing the expression pattern. (**G**) Heat maps of cerebellar brain tumor shared genes across the time points. (**H**) Venn diagram showing the number of genes that were specifically upregulated in the cerebellum at P7. (**I**) Heat map and gene list of cerebellar-specific P7 upregulated genes. (**J**) Dynamic Kif11 expression change in astrocytes between P7 and P14 (*n* = 4 per each group, **** *p* < 0.0001). Data are shown as the mean ± SEM.

## Data Availability

The datasets generated for this study can be found in the GEO repository, accession GSE252519 (access date: 30 June 2024). The source data are provided in this paper. All other data in this article are available from the corresponding author upon reasonable request.

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
