# Peer review of "Comparative Transcriptomic Analysis of Cerebellar Astrocytes across Developmental Stages and Brain Regions"

_ijms, 2024, doi:10.3390/ijms25021021_

Round 1
Reviewer 1 Report
Comments and Suggestions for Authors
This is a resourceful paper describing the developmental characteristics of cerebellar astrocyte transcriptomics. The comparison with brain cancer cells is intriguing. I don’t have anything objectionable. The below are vastly optional.
Line 20: temporal-specific … at specific time points ()
Line 40: potentials4
Fig 2E,G: The procedures to for generating these colored matrices should be explained in the methods section.
Fig 3B text: The explained variance should be described for PC1 and PC2 (i.e., the proportion of pc’s eigenvalue).
Supplementary tables should contain the numerical values used for the selection of the genes.
The data archive location for this study’s transcriptomics should be noted.
Line 252: delete “We”
Line 379: Since n=3 mice for each time, # male and female should be documented (or indicated in the figure).
Author Response
Reviewer responses and revision plan for “Comparative transcriptomic analysis of cerebellar astrocytes across developmental stages and brain regions” (Manuscript ID: ijms-2800197)
We thank the reviewers for their excellent critique and constructive feedback on our manuscript. To address these points we propose to conduct the following experiments, many of which are already underway. Below we provide a point-by-point response to each of the insightful comments provided by the reviewers. We feel that the reviewer comments and associated revisions have strengthened the clarity and rigor of our manuscript.
Reviewer #1
Summary/Comment 1-1: This is a resourceful paper describing the developmental characteristics of cerebellar astrocyte transcriptomics. The comparison with brain cancer cells is intriguing. I don’t have anything objectionable. The below are vastly optional.
Comment 1-2: Line 20: temporal-specific … at specific time points ()
Line 40: potentials4
Fig 2E,G: The procedures to for generating these colored matrices should be explained in the methods section.
Fig 3B text: The explained variance should be described for PC1 and PC2 (i.e., the proportion of pc’s eigenvalue).
Supplementary tables should contain the numerical values used for the selection of the genes.
The data archive location for this study’s transcriptomics should be noted.
Line 252: delete “We”
Line 379: Since n=3 mice for each time, # male and female should be documented (or indicated in the figure).
Response: We thank the reviewer for their overall positive assessment of our manuscript and appreciate the time they took to perform this evaluation. We have made all the requested edits to the manuscript. Many of these were corrected as part of the massive English editing that was suggested by Reviewer 2.

Reviewer 2 Report
Comments and Suggestions for Authors
In this manuscript, the authors comparatively analyzed changes in gene expression of astrocytes during the development of the cerebellum over time or by region within the brain single cell analysis techniques. In addition, the role of astrocytes in inferring the cause of pediatric brain cancer from cerebellum was suggested.
The RNA-seq analysis results in this manuscript well represent the expression patterns of cerebellum-specific genes, which were previously reported to play an important role in cerebellum development [1,2], and appear to have provided reliable results.
Since it provides very useful information, it seems appropriate to be published in this journal, IJMS, but many of the typos pointed out need to be corrected. Additionally, it would be good to add a more in-depth look into why the gene expression of cerebellum-specific astrocytes is different from that of other brain regions through the authors' research results.
1. Ha, T.J. et al., (2019) BMC genomics.
2. Jun, S. et al., (2023) Cell Reports.
Comments on the Quality of English Language
There are typos here and there so it doesn't look like it was edited well.
Please correct the parts indicated below.
Line 57: contribute -> contributes
Line 167, 405: gene -> genes
Line 170: No period
Line 252: Remove ‘We’
Line 288, tumor -> tumors
Line 412: 100mM -> 100 um
Author Response
We thank the reviewers for their excellent critique and constructive feedback on our manuscript. To address these points we propose to conduct the following experiments, many of which are already underway. Below we provide a point-by-point response to each of the insightful comments provided by the reviewers. We feel that the reviewer comments and associated revisions have strengthened the clarity and rigor of our manuscript.
Summary/Comment 2-1: In this manuscript, the authors comparatively analyzed changes in gene expression of astrocytes during the development of the cerebellum over time or by region within the brain single cell analysis techniques. In addition, the role of astrocytes in inferring the cause of pediatric brain cancer from cerebellum was suggested.
The RNA-seq analysis results in this manuscript well represent the expression patterns of cerebellum-specific genes, which were previously reported to play an important role in cerebellum development [1,2], and appear to have provided reliable results.
Since it provides very useful information, it seems appropriate to be published in this journal, IJMS, but many of the typos pointed out need to be corrected. Additionally, it would be good to add a more in-depth look into why the gene expression of cerebellum-specific astrocytes is different from that of other brain regions through the authors' research results.
1. Ha, T.J. et al., (2019) BMC genomics.
2. Jun, S. et al., (2023) Cell Reports.
Response: We appreciate the reviewers careful consideration of our results and overall positive critique of our manuscript. In order to expedite the publication of our manuscript, we have chosen not to include a deeper investigation into why gene expression profiles of cerebellar astrocytes are different from other regions. We agree that this is a potentially very important question, that could reveal new insights into regional heterogeneity of astrocytes, however such analysis (in our humble opinion) is beyond the scope of this current study.
Comment 2-2: There are typos here and there so it doesn't look like it was edited well.
Response: This is a gross oversight on our part and we profusely apologize for these mistakes. In our defense we had this paper edited by a local scientific editor, who was recently hired by our department. Clearly, we should have spent more time editing the editor.
In the revised manuscript, I (corresponding author) edited the manuscript myself and it should now be acceptable. Again, I personally apologize for this embarrassing oversight.
Comment 2-3: Please correct the parts indicated below.
Line 57: contribute -> contributes
Line 167, 405: gene -> genes
Line 170: No period
Line 252: Remove ‘We’
Line 288, tumor -> tumors
Line 412: 100mM -> 100 um
Response: We have made all these corrections.
